# Transfer of skin microbiota between two dissimilar autologous microenvironments: A pilot study

Benji Perin[1]*, Amin Addetia[2¤], Xuan Qin[2,3]

**1** University of Washington Division of Dermatology and Dermatology Residency, Seattle, WA, United States of America, **2** Seattle Children's Hospital, Seattle, WA, United States of America, **3** University of Washington Department of Laboratory Medicine, Seattle, WA, United States of America

¤ Current address: University of Washington Department of Laboratory Medicine, Seattle, WA, United States of America

* bip2@uw.edu

**Data Availability Statement:** Fastq files are deposited in the NCBI Sequence Read Archive under the BioProject Accession Number PRJNA523412. https://www.ncbi.nlm.nih.gov/bioproject.

## Abstract

Dysbiosis of skin microbiota is associated with several inflammatory skin conditions, including atopic dermatitis, acne, and hidradenitis suppurativa. There is a surge of interest by clinicians and the lay public to explore targeted bacteriotherapy to treat these dermatologic conditions. To date, skin microbiota transplantation studies have focused on moving single, enriched strains of bacteria to target sites rather than a whole community. In this prospective pilot study, we examined the feasibility of transferring unenriched skin microbiota communities between two anatomical sites of the same host. We enrolled four healthy volunteers (median age: 28 [range: 24, 36] years; 2 [50%] female) who underwent collection and transfer of skin microbiota from the forearm to the back unidirectionally. Using culture methods and 16S rRNA V1-V3 deep sequencing, we compared baseline and mixed ("transplant") communities, at T = 0 and T = 24 hours. Our ability to detect movement from one site to the other relied on the inherent diversity of the microenvironment of the antecubital fossa relative to the less diverse back. Comparing bacterial species present in the arm and mixed ("transplant") communities that were absent from the baseline back, we saw evidence of transfer of a partial DNA signature; our methods limit conclusions regarding the viability of transferred organisms. We conclude that unenriched transfer of whole cutaneous microbiota is challenging, but our simple technique, intended to move viable skin organisms from one site to another, is worthy of further investigation.

## Introduction

Until recently, skin microbiota research has been primarily descriptive. Foundational studies in healthy subjects have revealed remarkable topographical diversity [1] and temporal stability [2]. Increasingly, we are recognizing associations between microbial dysbiosis and inflammatory skin conditions. Most clearly elucidated with the role of *Staphylococcus aureus* in atopic

**Funding:** BP received funding by The National Center for Advancing Translational Sciences of the National Institutes of Health (https://ncats.nih.gov, Award Number UL1TR000423), as awarded by the Seattle Children's CCTR Pediatric Pilot Fund Program. The funders had no role in study design, data collection and analysis, decision to publish, or preparation of the manuscript.

**Competing interests:** The authors have declared that no competing interests exist.

dermatitis [3,4], important microbial trends of dysbiosis are also emerging in acne [5,6] and hidradenitis suppurativa [7], among other conditions.

The clinical promise of transferring microbiota has been demonstrated with fecal microbiota transplantation, which has shown curative potential on the individual level (*C. difficile* colitis) in addition to its benefit to the greater biosphere with enhanced antimicrobial stewardship [8,9]. In the emerging field of cutaneous bacteriotherapy, studies have focused on applying a single species to target sites to treat atopic dermatitis, given these species' ability to inhibit *Staphylococcus aureus* growth [10–12].

No studies to date have explored the feasibility of performing a skin microbiota transplant that moves the entire cutaneous bacterial community, with its complex web of metabolic interactions. The mechanistic significance of transferring a community rests upon the fact that many microbes need their community partner, ie some microbes make associations of obligately mutualistic metabolism, sometimes termed syntrophy, or cross-feeding mode of living [13]. In humans, research in this area has focused on pathogens that evolve co-dependent isogenic variants, acting like a multicellular organism to produce functional antibiotic resistance [14,15] However, in human gut microbiome research, there is emerging evidence of cross-feeding of commensal bacteria to produce bioactive short chain fatty acids in the healthy host [16–18]. Given this growing body of evidence for syntrophy in microbial systems of the healthy human host, we believe that transferring the naive microbial community without species bias introduced by an enrichment step *in vitro*, is a valid investigational approach for the treatment of inflammatory skin disease.

Within this context, our study asks whether moving superficial cutaneous microbial communities is feasible. Our experimental design relies on the topographical variation of skin microbiota within a single host. We selected sites with a contrasting composition of microbes, the antecubital fossa and the upper back [1]. Using both sequencing and traditional microbiological culture, we took the advantage of the differences in baseline populations to distinguish a signal of successful transfer. Here, we aim to follow the signal of these transferred species and demonstrate that a simple and inexpensive method for moving superficial skin microbiota can create a viable and representative transplant.

## Materials and methods

The study was approved by Seattle Children's Institutional Review Board. Written consent was obtained for study participants. The study was conducted at Seattle Children's Hospital from January-March 2017.

### Recruitment of study participants

Healthy medical students 23–37 years of age were recruited for the study from the University of Washington School of Medicine, screened with exclusion criteria by questionnaire, and consented at the time of the screening swab. Exclusion criteria were no antibiotics in the last six months; generally healthy; no skin disease other than acne, keratosis pilaris, or dry skin; no soaping/scrubbing of arms and back when bathing; no bathing with antibacterial soap.

Because our preliminary trials revealed that skin microbiota biomass varies considerably between individuals, volunteers' antecubital fossae were screened for a minimum bioburden.

To assess bioburden, a moistened swab (BD, ESwab) with 0.85% sterile saline; (Remel) was vigorously rubbed on a 2cm x 2 cm area of antecubital fossa. This is the same saline we use throughout the experiment, including for collection of baseline samples, collecting bacteria for transfer pellet, and recovering the transferred pellet. The swab was placed in 1 mL of modified liquid Amies medium (BD) and vortexed for 30 seconds. A blood agar (BA) plate (Remel) was

inoculated with 0.1 mL of the Amies medium and incubated aerobically at 35˚C for 48 hours. We calculated cutaneous biomass and evaluated each volunteer's bioburden. We set a limit of >1000 colony forming units per milliliter Amies medium (CFU/mL) for inclusion criteria.

Using cutaneous bacterial biomass as inclusion criteria ensured there was sufficient bioburden for our subsequent analyses. We screened nine volunteers, all of whom gave written informed consent. Of them, two men and two women (median age: 28 [range: 24, 36]) had sufficient biomass for inclusion. The individual in this manuscript (identifiable in S1 Photo) has given written informed consent (as outlined in PLOS consent form) to publish these case details.

## Collection of baseline samples

Study participants did not bathe for at least 24 hours prior to sampling. On the day of sampling, the subject's arms and back were fitted with pre-constructed, raised grids of waterproof medical tape (Nexcare Absolute Waterproof, 3M; S1 Photo; Fig 1). Baseline samples (*Ba*, *Bb*; Fig 1) from the arms (*Ba*) and back (*Bb*) were obtained by vigorously rubbing the designated 2.5 cm x 3.0 cm grid-squares for 30 seconds with dampened swabs. For all adjacent samples, swabs of one grid-square in went in 1 mL Amies for culture, and the other grid-square in 0.5 mL of PowerBead solution (Qiagen) for PCR. Culture and PCR methods are outlined in detail in the following sections.

## Moving the arm microbiota to the back

To create the bacterial transfer pellets, the donor sites (*D*; Fig 1) were vigorously rubbed with dampened swabs. We then submerged each swab in 1 mL saline and vortexed for 30 seconds. Next, we transferred the saline to a DNA-free microcentrifuge tube and centrifuged at 2,000 x g for 5 minutes, followed by a second, equivalent centrifugation with the tube rotated 180 degrees [19]. This created a pellet in the apex of the tube. We removed all but 50 μL of supernatant, and resuspended the pellet in the remaining supernatant, creating a solution with the consistency of thick mucus. This solution was pipetted directly onto the appropriate recipient site ($T_0$, $T_{24}$; Fig 1), and spread with a disposable inoculating loop (Fisherbrand). There was no pre-treatment of the recipient sites prior to transfer.

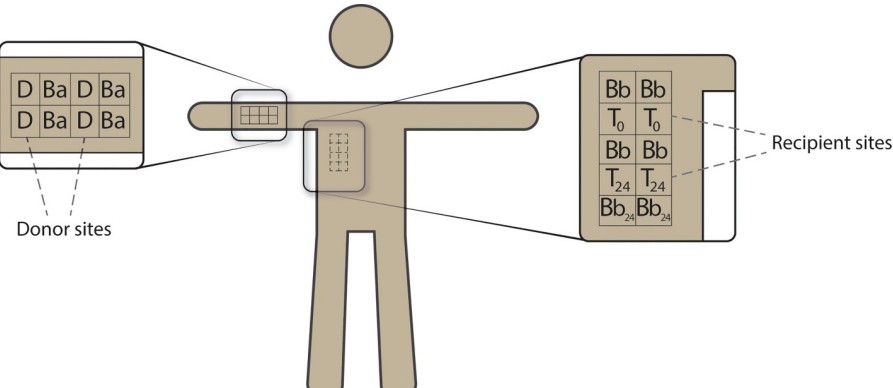

**Fig 1. Overview of sites for one replicate of the experiment (one replicate equals one anatomic "side" of a study subject, here right arm and back).** For each pair of adjacent samples, one is cultured, one is sequenced. [Ba] Baseline samples of arm at T = 0; [Bb] Baseline samples of back at T = 0; [D] donor sites for generation of bacterial pellet (transplant); [T0 ] T = 0 samples of recipient sites for bacterial pellet (transplant) mixed with back microbiota; [Bb24 ] baseline samples of back at T = 24; [T24 ] T = 24 samples of recipient sites for bacterial pellet (transplant) mixed with back microbiota.

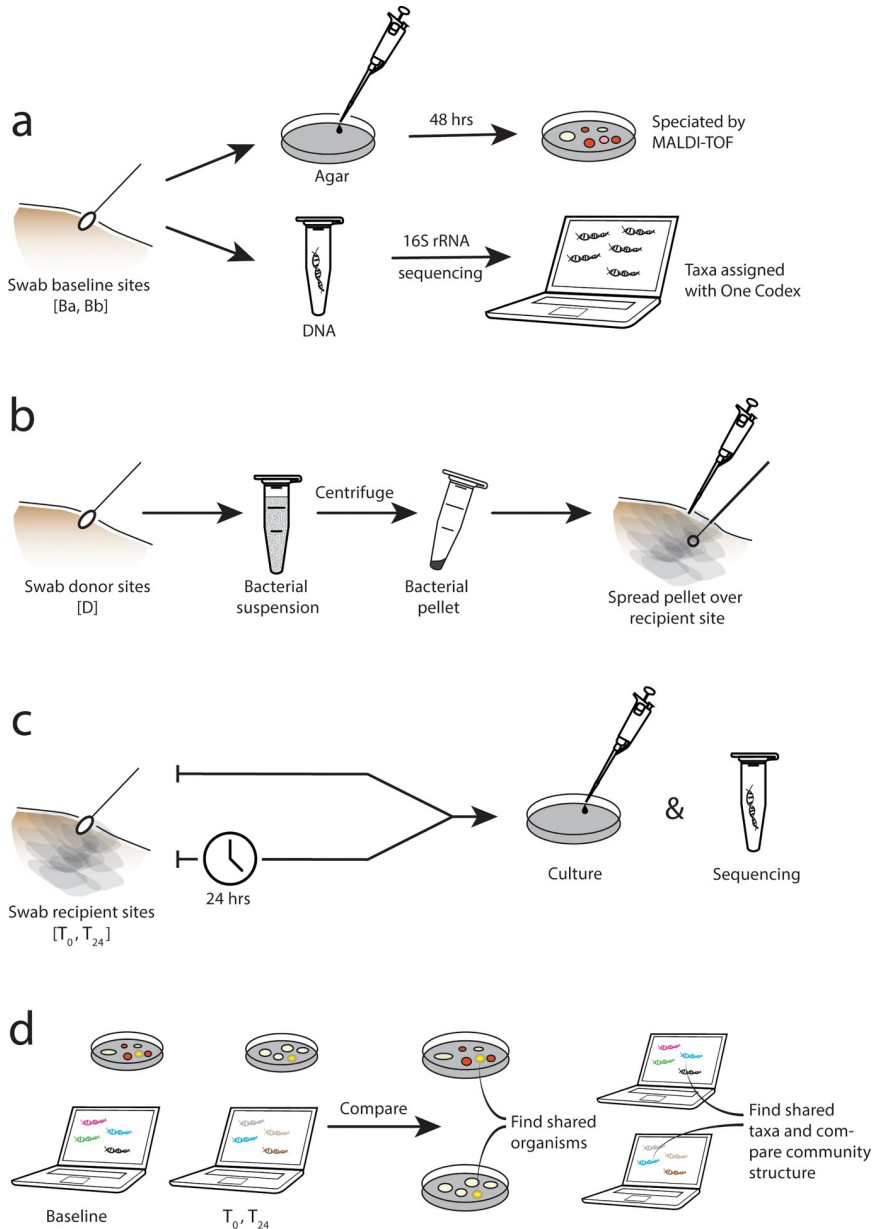

**Fig 2. Study overview—methods and analysis.** [A] Baseline samples collected from arm and back; [B] making and transferring the bacterial pellet (transplant); [C] sampling of recipient sites comprised of bacterial pellet mixed with resident back microbiota at T = 0 and T = 24 hours; [D] comparison of $T_0$, $T_{24}$ (mixed) sites to baseline sites (looking for evidence of cultured organisms and sequenced taxa that exist in the baseline arm and $T_0$, $T_{24}$ samples, but are absent in baseline back samples, which serve as controls).

### Assessing the efficacy of our microbiota transfer technique

To assess the efficacy of our technique, we collected transferred pellet samples immediately and 24 hours after we spread the pellet across the recipient sites ($T_0$, $T_{24}$; Fig 1). The $T_0$ samples were collected with the same method used for obtaining the baseline samples as described above (Fig 2).

After 24 hours, we recreated the tape grids in exactly the same position on the subject's back (marked on day one with surgical pen). Study subjects were instructed not to bath

between placement and harvest of the bacterial pellet. We then collected the transferred pellet samples ($T_{24}$; Fig 1) and baseline back samples ($Bb_{24}$; Fig 1). All the T = 0 and T = 24 samples were analyzed by both bacterial culture and 16S rRNA deep sequencing (Fig 2).

In total, there were eight replicates of the entire experiment: one on each anatomical side of the four participants (one replicate being right arm + right upper back; second replicate being left arm + left upper back). For every replicate, culture and 16S deep sequencing each owned an adjacent grid-square at each time point.

## Analyzing microbiota composition with 16S rRNA sequencing

The swabs were placed into 0.5 mL of PowerBead solution (Qiagen) and vortexed for 30 seconds. The samples were transferred to bead tubes provided with the DNeasy PowerSoil Kit (Qiagen), and 0.06 mL of C1 solution was added to each tube. The tubes were briefly vortexed and incubated at 70°C for 10 minutes. The samples were lysed with a Precellys24 (Bertin Technologies) operated at 5000 RPM for 30 seconds. The manufacturer's instructions were followed for the remaining extraction and purification steps.

A negative (reagent-only) control and a positive control of five organisms–*Candida albicans* ATCC 10231, *Staphylococcus aureus* ATCC 29213, *Streptococcus pneumoniae* ATCC 49619, *Pseudomonas aeruginosa* ATCC 27853 and *Haemophilus influenza* ATCC 49247 –were included with each set of extractions. Negative environmental control swabs (swabs that were opened and exposed to the air of the sampling room for about 15 seconds) were collected for each subject (both at T = 0 and T = 24) and extracted concurrently with the experimental swabs.

All amplification and deep sequencing was completed by the University of Minnesota Genomics Center (UMGC), with the V1-V3 region of the 16S rRNA gene amplified using the UMGC dual-indexing protocol, as previously described [20]. Sequencing was completed on the Illumina MiSeq using the 300 base pair, paired end approach.

Fastq files were uploaded to One Codex [21] and taxa assigned according to the targeted loci database (closed reference). The read counts for each sample were analyzed using Calypso v8.20 [22], without read filter or removal of rare taxa, using total sum normalization without transformation, and the Greengenes taxonomy database (v13.8). Shannon Index was used for beta diversity analysis, and PCoA plot with Bray-Curtis index for comparing community structure.

## Analyzing microbiota composition with traditional culture methods

Swabs were placed in 1 mL of modified liquid Amies medium and vortexed for 30 seconds. A BA plate, mannitol salt agar (MSA) plate (BD) and phenylethyl alcohol agar (PEA) plate (Remel) were each inoculated with 0.1mL of Amies medium. An additional BA plate was inoculated with 0.1 mL of a 1:10 dilution of Amies medium and a third BA plate was inoculated with 0.1 mL of a 1:100 dilution of Amies medium. A 2 mL aliquot of Reasoner's 2A (R2A) broth (Teknova) containing a vancomycin disk (30 μg, BD) and 0.05 mL of amphotericin B (250 μg/mL, Fisher) [23] was inoculated with 0.5 mL of Amies media.

The BA and MSA plates were incubated aerobically at 35°C for 48 hours and screened for growth. Each unique morphotype was subcultured to a BA plate and identified via matrix-assisted laser desorption/ionization time of flight mass spectrometry (MALDI-TOF MS, Bruker Daltonics Inc.). Colony counts, measured in CFU/mL, were obtained for each morphotype. The PEA plate was placed in a sealed box with an AnaeroPack System (MGC) and incubated at 35°C for 120 hours. The R2A broth was incubated at 32°C at a constant shaking of 150 RPM for 48 hours. A BA plate was inoculated with 0.2 mL of R2A broth and incubated aerobically at 35°C for 48 hours. As with the BA plates, for the PEA and R2A-inoculated plates each unique morphotype was identified via MALDI-TOF MS.

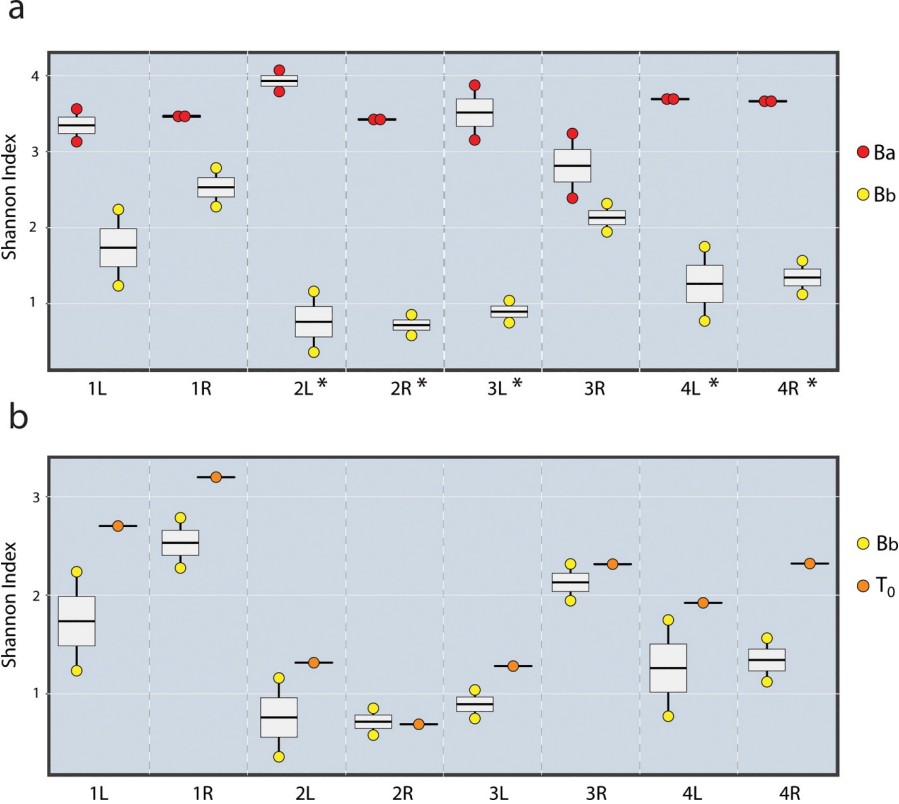

**Fig 3. Boxplot of Shannon Diversity Index for each replicate of the experiment: subjects 1–4, [R]ight and [L]eft side.** Shannon Diversity weighs both the number of different species and their relative abundance in the sample. Here we compare (a) baseline arm [Ba] samples to baseline back samples [Bb], and (b) Bb samples to recipient sites for bacterial pellet at T = 0 [$T_0$]. Significant difference ($p<0.05$) by ANOVA analysis is denoted with a (*). There was no significant difference nor trend comparing $Bb_{24}$ and $T_{24}$ samples; for this reason they are not included here.

We classified two bacterial isolates as the "same" only when both the MALDI identification and the pattern of morphological properties (by size, shape, pigment, texture, etc.) of the two organisms were identical. Colony morphologies and MALDI identifications were compared between plates grown from all sites on the same side of each study participant's body (plates compared within each replicate of the experiment). Use of colony morphology to identify different species is a common tool in microbiology; colony morphology has also been shown to distinguish different strains of the same bacterial species [24].

## Results

At baseline with the 16S deep sequencing data, we found the microbial community of the antecubital fossa (*Ba*) was more diverse than the back (*Bb*) in all four subjects. This is reflected by the number of distinct species found at each site (median: 232 species unique to *Ba* [range: 120, 363]; 57 unique to *Bb* [28, 103]; and 155 shared between *Ba* and *Bb* [123, 252]) and also in the increased Shannon diversity of the arm as compared to the back (significant in 5/8 replicates) (Fig 3A).

Comparisons of relative abundance of bacteria in the arm and back samples also demonstrate the differences in microbial signature between the two anatomical sites (Fig 4). Although *Cutibacterium* accounted for the majority of reads in most back samples, this was not true for the much more diverse antecubital fossae. Fig 4 also demonstrates that while antecubital fossae

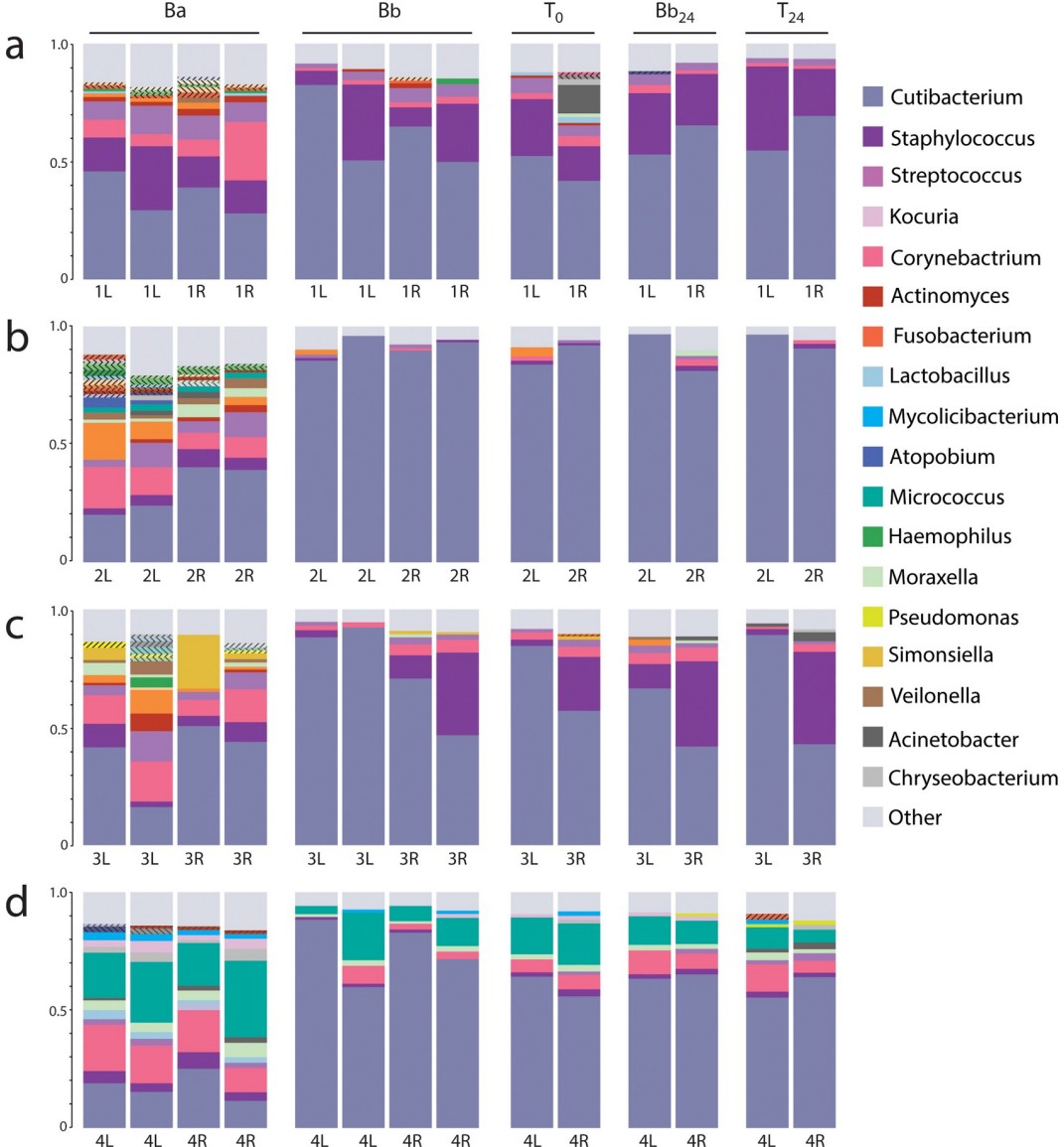

**Fig 4. Relative readcount by genus (% of classified reads).** (a) Subject 1, (b) Subject 2, (c) Subject 3, (d) Subject 4. While the differences between the baseline arm [Ba] and back [Bb] are striking at this resolution, evidence of successful movement of arm bacteria is more difficult to discern in the samples of recipient sites [$T_0$, $T_{24}$]. The 18 most common genera (>3% total reads in a sample) are labelled with a corresponding color. Genera with between 1% and 3% total reads have their own color but are not labelled in the key; these species are marked with diagonal lines to distinguish them from those in the color key. Genera with <1% are grouped in "other". On average, 96% of reads were classified in each sample (range: 91% - 100%).

across individuals show commonalities (*Staphylococcus*, *Streptococcus*, and *Corynebactrium* species playing prominent roles along with *Cutibacterium*), we also see differences between study subjects that set them easily apart. Subject 2 hosts notable quantities of *Fusobacterium*; subject 3, *Simonsiela* (commonly found in the oral cavity of dogs); and subject 4, *Micrococcus*. These differences form the basis for the growing field of microbiome forensics [25].

We saw unique, transferred, arm species (absent in *Bb* samples) appear in all $T_0$ and $T_{24}$ samples. By sequencing, a median of 34 arm-only species [range: 18,85] appeared in the $T_0$ samples, with a median of 4 arm-only species [range: 1,16] persisting in the $T_{24}$ samples. The

most common of these organisms were *Gardnerella vaginalis*, *Brachybacterium faecium*, *Janithobacterium lividum*, and unclassified species of *Actinomyces*, *Anaerococcus*, *Microbacteriaceae*, and *Dermabacteriaceae* (Table 1). By culture, we also saw a limited number of bacteria unique to the arm (absent in *Bb* samples) appear in $T_0$ and $T_{24}$ samples (Table 2). Our difficulty in identifying the movement of unique live bacteria through culture techniques is best appreciated in supplementary data (S1 Dataset), which details the >900 subtyped colonies from our four study subjects. These data show that the majority of species which we grew were the *Staphylococcus*, *Streptococcus*, *Corynebacteria*, and *Roseamonas* that reside at baseline on both the arm and the back. Despite our attempts to incubate in R2A with vancomycin to inhibit overgrowth of *Staphylococcus*, we were unable to cultivate the rare, primarily gram negative species that are unique to the arm. Nevertheless, three arm-only species in $T_0$ samples were identified by both sequencing *and* culture (in bold italics in Tables 1 and 2). Although very limited in number, these three species offer some support for the movement not only of DNA but viable organisms. Further evidence for the movement of viable organisms are the unique colony morphotypes of species common to both sites that we demonstrated moving from the arm to the back in our study subjects (Table 2).

Besides identifying specific "arm" bacterial DNA moved to the back, we assessed the transfer of DNA signature by comparing community compositions with diversity analysis and PCoA Bray-Curtis plot. The $T_0$ samples were more diverse than *Bb* samples in 7 of 8 replicates, although this trend was not significant (Fig 3B). Also, in a projection of community composition (PCoA Bray-Curtis plot), five of eight $T_0$ samples shift towards the *Ba* cluster and away from the *Bb* cluster (Fig 5). Specifically, three of the $T_0$ samples plot between their baseline back samples (this is what we would expect if the $T_0$ samples were not impacted by the community composition of the transfer pellet). Five of the $T_0$ samples have moved to the right of both of their respective back samples, towards the arm samples, showing a qualitative impact of the transfer pellet on the structure of the community.

Not all bacterial DNA from the arm was moved to the back with our transplant process. The sequencing data show a median of 16% of unique arm bacterial species were recovered from $T_0$ samples [range: 10% - 25%]. This result shows our incomplete success in moving the entire arm skin microbiota DNA signature.

Our positive controls (one with each of four DNA extraction runs) were consistent with each other and showed that *Staphylococcus aureus* was underrepresented in our final results, either because of incomplete extraction of DNA or because of bias in the PCR-sequencing pipeline. We were reassured by the result of negative controls (environmental and reagent), which showed read counts ten times lower than experimental samples. As expected, negative reagent controls showed read counts for only a limited number of species.

We include here one further result from preparatory trials for our study, a simple measurement of whether the process of pelleting the bacteria by centrifugation (the process of preparing bacteria for transfer) resulted in loss of viability. From two adjacent sites (of equal surface area) of the antecubital fossa, we saw equivalent growth on blood agar from bacterial pellets (created with the centrifuge technique, as described above in Materials and Methods, and resuspended in Amies solution) and baseline swabs (mixed directly into an equivalent volume of Amies solution) (Table 3).

## Discussion

Current investigations in skin microbiota transplantation show promise in the application of single strains of bacteria to lesional skin. Myles, et. al. showed that certain Gram-negative species, particularly *Roseomonas mucosa* collected from the skin of healthy volunteers, have

**Table 1. List of species identified by *sequencing* that were present in the baseline arm [Ba], absent in baseline back [Bb], and present in the recipient site sample [T₀].**

**Subjects 1 and 2; left [L] and right [R] side**

| 1L | 1R | 2L | 2R |
|---|---|---|---|
| Actinomyces sp.* | Agathobaculum butyriciproducens | Gardnerella vaginalis | Brachybacterium faecium |
| Anaerococcus unclassified | Atopobium parvulum | Janthinobacterium lividum | Gardnerella vaginalis |
| Brachybacterium faecium | Oxalobacteraceae unclassified | Microbacteriaceae unclassified | Actinomyces turicensis |
| Janthinobacterium lividum | Peptoniphilus indolicus | Alphaproteobacteria unclassified | Eggerthella sinensis |
| Microbacteriaceae unclassified | Pseudomonas fluorescens group unclassified | Candidatus Peptoniphilus massiliensis | Enterobacter ludwigii |
| Actinomyces odontolyticus* | Pseudomonas synxantha | Dialister propionicifaciens | Gordonibacter pamelaeae |
| Agathobaculum butyriciproducens | Sphingomonas melonis | Eggerthella sinensis | Intrasporangiaceae unclassified |
| Alphaproteobacteria unclassified | Acinetobacter haemolyticus | Firmicutes unclassified | Moraxella unclassified |
| Betaproteobacteria unclassified | Arsenicicoccus bolidensis | Gordonibacter pamelaeae | Prevotella veroralis |
| Brevundimonas nasdae | Arthrobacter sp. | Moraxella unclassified | Sphingomonas melonis |
| Flavobacteriaceae unclassified | Blastococcus aggregatus | Oxalobacteraceae unclassified | Acinetobacter unclassified |
| Lactobacillus jensenii | Candidatus Microthrix calida | Pseudomonas unclassified | Bacillus sp. N6 |
| Lysobacter unclassified | Chryseobacterium halperniae | Rhizobiales unclassified | Chitinophagaceae unclassified |
| Micrococcus unclassified | Chryseobacterium indologenes | Roseomonas mucosa | Corynebacterium confusum |
| Peptostreptococcus anaerobius | Clostridiales Family XIII. | Simonsiella muelleri | Corynebacterium matruchotii |
| Pseudomonas synxantha | Incertae Sedis unclassified | Triticum aestivum | Fusobacterium nucleatum* |
| Serratia liquefaciens | Eikenella corrodens | Actinomycetaceae unclassified | Microbacterium esteraromaticum |
| [Clostridium] saccharolyticum | Janibacter sanguinis | Amycolatopsis orientalis | Mycobacterium asiaticum |
| Anaerococcus prevotii | Leptotrichia goodfellowii | BOP clade unclassified | Pseudomonas fluorescens |
| Atopobiaceae unclassified | Mobiluncus curtisii | Corynebacterium minutissimum* | Rhizobiaceae unclassified |
| Campylobacter gracilis | Mogibacterium unclassified | Delftia unclassified* | Sphingomonadaceae unclassified |
| Capnocytophaga granulosa | Ottowia beijingensis | Dialister unclassified | Streptococcus cristatus |
| Chryseobacterium lathyri* | Peptoanaerobacter stomatis | Flaviflexus salsibiostraticola | |
| Citrobacter freundii | Porphyromonas endodontalis | Gordonia unclassified | |
| Collinsella aerofaciens | Prevotella micans | Lactobacillus acetotolerans | |
| Coprococcus eutactus | Prevotella timonensis* | Massilia aurea* | |
| Cupriavidus metallidurans* | Rhizobium unclassified | Massilia unclassified | |
| Deinococcus unclassified | Sphingomonas phyllosphaerae* | Negativicutes unclassified | |
| Dermacoccus unclassified | Streptococcus pneumoniae | Paraeggerthella hongkongensis | |
| Dialister pneumosintes | Treponema vincentii* | Peptoniphilus asaccharolyticus* | |
| Dysgonomonas mossii | Varibaculum anthropi | Peptoniphilus lacrimalis | |
| Enterobacteriaceae unclassified | Varibaculum cambriense | Rhodococcus erythropolis | |
| Gammaproteobacteria unclassified* | | Rothia mucilaginosa | |
| Geobacillus stearothermophilus | | Sphingobium yanoikuyae | |
| Ileibacterium massiliense | | Streptomyces chungwhensis | |
| Libanicoccus massiliensis | | | |
| Luteolibacter unclassified | | | |
| Microbacterium oxydans | | | |
| Ottowia unclassified | | | |
| Parvimonas unclassified | | | |
| Peptococcus sp. feline oral taxon 012 | | | |
| Prevotella melaninogenica | | | |
| Prevotella shahii | | | |
| Prevotella sp. oral taxon 292 | | | |

*(Continued)*

**Table 1.** (Continued)

| Pseudoclavibacter alba | | | |
|---|---|---|---|
| Rothia unclassified | | | |
| Solirubrobacter ginsenosidimutans | | | |
| Sphingobium xenophagum | | | |
| Staphylococcus hominis | | | |
| Xanthomonadaceae unclassified | | | |
| Xanthomonas albilineans | | | |

**Subjects 3 and 4; left [L] and right [R] side**

| 3L | 3R | 4L | 4R |
|---|---|---|---|
| Actinomyces sp. | Actinomyces sp. | Anaerococcus unclassified | Anaerococcus unclassified |
| Brachybacterium faecium | Dermabacteraceae unclassified* | Dermabacteraceae unclassified | Janthinobacterium lividum |
| Dermabacteraceae unclassified | Gardnerella vaginalis* | Actinomyces odontolyticus* | Anaerococcus hydrogenalis |
| Microbacteriaceae unclassified | Actinomyces neuii | Actinomyces turicensis | Bacillales unclassified |
| *Actinomyces neuii* | Atopobium parvulum | Anaerococcus hydrogenalis | Brevundimonas vesicularis |
| Bacillales unclassified | Prevotella veroralis | Betaproteobacteria unclassified | Candidatus Peptoniphilus massiliensis |
| Enterobacterales unclassified | Rhizobiales unclassified | Brevundimonas nasdae | Corynebacterium mucifaciens |
| Flavobacteriaceae unclassified | Arabidopsis thaliana* | Brevundimonas vesicularis | Enterobacter ludwigii |
| Helcobacillus massiliensis | Corynebacterium macginleyi | *Corynebacterium mucifaciens* | Enterobacterales unclassified |
| Mesangiospermae unclassified | Glutamicibacter ardleyensis | Dialister propionicifaciens | Firmicutes unclassified |
| Micrococcus unclassified | Hydrogenophilus islandicus | Friedmanniella spumicola | Friedmanniella spumicola |
| Peptostreptococcus anaerobius | Lachnospiraceae unclassified | Helcobacillus massiliensis | Intrasporangiaceae unclassified |
| Streptococcus parasanguinis | Lactobacillus delbrueckii | Lactobacillus gasseri | Lactobacillus gasseri |
| Triticum aestivum | Leuconostoc garlicum | Lactobacillus jensenii | Macrococcus equipercicus |
| Actinomyces oris | Microbacterium paraoxydans | Lysobacter unclassified | Methylobacterium unclassified |
| Bergeyella cardium | Micrococcus luteus | Macrococcus equipercicus | Mycolicibacterium iranicum |
| Bergeyella unclassified | Nesterenkonia halotolerans | Mesangiospermae unclassified | Neisseria unclassified |
| Brachybacterium unclassified* | Roseomonas riguiloci | Methylobacterium unclassified | Rhodobacteraceae unclassified |
| Campylobacter concisus | | Mycolicibacterium iranicum | Sphingomonas desiccabilis* |
| Chryseobacterium hominis | | Neisseria unclassified* | Staphylococcus haemolyticus |
| Chryseobacterium unclassified | | Peptoniphilus indolicus | Actinomyces mediterranea |
| Corynebacterium accolens | | Pseudomonas fluorescens | Amaricoccus macauensis |
| Gemella sanguinis | | group unclassified | Burkholderiales Genera incertae sedis unclassified |
| Microbacterium unclassified | | Pseudomonas unclassified | Caulobacter vibrioides* |
| Parvimonas micra | | Rhodobacteraceae unclassified* | Devosia neptuniae |
| Parvimonas sp. oral taxon 110 | | *Roseomonas mucosa* | Gemella haemolysans |
| Pentapetalae unclassified | | Serratia liquefaciens | Gemmobacter caeni* |
| Poaceae unclassified | | Simonsiella muelleri* | Granulicatella para-adiacens |
| Prevotella histicola | | Sphingomonas desiccabilis | Janibacter unclassified |
| Prevotella salivae | | Staphylococcus haemolyticus | Lactobacillus reuteri* |
| Pseudogracilibacillus auburnensis | | Streptococcus parasanguinis | Leptotrichia trevisanii |
| Stenotrophomonas maltophilia* | | Acinetobacter septicus | Luteimonas unclassified |
| | | Agrobacterium fabrum* | Macrococcus canis |
| | | Agrobacterium tumefaciens | Macrococcus unclassified |
| | | Altererythrobacter salegens | Mesorhizobium loti |
| | | Aridibacter kavangonensis | Methylobacterium radiotolerans |
| | | Blastocatellaceae unclassified | Methylosinus trichosporium |
| | | Brachybacterium conglomeratum | |

*(Continued)*

**Table 1.** (Continued)

| | | | |
|---|---|---|---|
| | | Brevundimonas unclassified* | Microbacterium saccharophilum |
| | | Burkholderiaceae unclassified | Micropruina glycogenica |
| | | Burkholderiales unclassified | Mycolicibacterium austroafricanum* |
| | | Caulobacteraceae unclassified | Nakamurella sp. |
| | | Chryseobacterium gleum | Neisseria meningitidis |
| | | Chryseobacterium hispanicum | Nioella sediminis |
| | | Chryseobacterium taiwanense* | Paraburkholderia tropica |
| | | Clostridiales unclassified | Paracoccus siganidrum |
| | | Deinococcus sp. | Paracoccus yeei |
| | | Dermacoccus nishinomiyaensis | Peptoniphilus coxii |
| | | Dietzia maris | Porphyromonas bennonis* |
| | | Fenollaria massiliensis | Roseomonas gilardii |
| | | Gordonia sputi | Sphingomonas echinoides |
| | | Granulicatella elegans | Staphylococcus equorum* |
| | | Haemophilus influenzae | Staphylococcus saprophyticus |
| | | Kouleothrix aurantiaca | Stenotrophomonas rhizophila |
| | | Lactobacillus johnsonii | Streptococcus oralis* |
| | | Massilia alkalitolerans | Streptococcus salivarius |
| | | Methylorubrum extorquens* | Veillonella parvula |
| | | Nakamurella multipartita | Vicinamibacter silvestris |
| | | Neisseria flavescens | |
| | | Neorhizobium huautlense | |
| | | Nocardiaceae unclassified* | |
| | | Nocardioides oleivorans | |
| | | Nocardioides sp. | |
| | | Nocardioides unclassified* | |
| | | Nonspecific* | |
| | | Oryza sativa | |
| | | Pantoea agglomerans* | |
| | | Pantoea vagans | |
| | | Paracoccus marinus* | |
| | | Paracoccus versutus | |
| | | Phenylobacterium unclassified | |
| | | Propionibacteriaceae unclassified | |
| | | Proteobacteria unclassified* | |
| | | Pseudomonas putida* | |
| | | Pseudomonas stutzeri | |
| | | Riemerella anatipestifer | |
| | | Sphingobacterium sp. enrichment culture clone* | |
| | | Sphingobium unclassified | |
| | | Sphingomonadales unclassified | |
| | | Sphingomonas guangdongensis | |
| | | Sphingomonas hengshuiensis | |
| | | Variovorax paradoxus | |
| | | Xanthomonadales unclassified | |
| | | Xanthomonas axonopodis | |
| | | Zhizhongheella caldifontis | |

(*Continued*)

**Table 1.** (Continued)

| | | Zoogloea oryzae | |
|---|---|---|---|

Species listed in blue cells occur in >2 replicates, species listed in orange cells occur in >1 replicates, and species listed in white boxes occur only once across replicates. Species in **bold italic** are examples where the culture data (derived from a sample taken centimeters away on the same individual) corroborates the sequencing data (present in the baseline arm [Ba], absent in baseline back [Bb], and present in the recipient site sample [$T_0$]). Species from $T_0$ that persist in the $T_{24}$ site (and remain absent at $Bb_{24}$ site) are annotated with a (*).

antimicrobial activity against *Staphylococcus aureus* [11], and in a phase 2 clinical trial, application of *R. mucosa* to active atopic dermatitis was associated with decreased disease severity, topical steroid requirement, and *S. aureus* burden [12]. Similarly, Gallo and Nakatsuji identified *Staphylococcus epidermidis* strains with antimicrobial activity against *S. aureus* [10]. In animal models of atopic dermatitis, the application of Nakatsuji's *S. epidermidis* eliminated *S. aureus* colonization. In the context of these studies, our investigation reflects a slightly different goal: to move interconnected *communities* of microbes, with their web of metabolic interactions, from healthy individuals to the skin of patients with inflammatory skin disease.

Tables 1 and 2 summarize our evidence supporting the feasibility of transferring a partial DNA signature from one site to another, listing species that were present in the baseline arm [*Ba*], absent in baseline back [*Bb*], and were recovered from recipient sites [$T_0$]. These unique-to-arm species likely represent the tip of a larger transplant iceberg, i.e. they could serve as a proxy for the majority of successfully transferred organisms that are species shared between the two sites, and which we could not detect with 16S sequencing. We also interpret the shift of community structure between *Bb* and $T_0$ as evidence that our intervention made the recipient back sites more "armlike" in their community composition (Figs 3B and 5).

Despite the viability of the pelleted bacteria in trials and our success in growing some unique arm organisms from $T_0$ samples, our results most clearly show the movement of DNA, with only limited corroboration that the DNA is recovered from live organisms. We explore this limitation, and how future studies can better assess the viability of transferred bacteria in "Limitations", below.

While the DNA of several of the unique, rare arm bacteria persisted at 24 hours in their new back environment, we saw a steep drop in this signal. Given their new microenvironment we cannot say what dynamics led to the failure of these bacteria to colonize the recipient site. If

**Table 2. List of unique morphotypes of species identified by *culture* and MALDI-TOF that were present in the baseline arm [Ba], absent in baseline back [Bb], and present in the recipient site sample [$T_0$].**

| Subjects 1 and 2; left [L] and right [R] side | | | |
|---|---|---|---|
| **1L** | **1R** | **2L** | **2R** |
| None | Staphylococcus epidermidis | Micrococcus luteus | Staphylococcus capitis |
| | Staphylococcus sp[1] | | Micrococcus luteus |
| **Subjects 3 and 4; left [L] and right [R] side** | | | |
| **3L** | **3R** | **4L** | **4R** |
| Staphylococcus epidermidis | Staphylococcus sp[1] | ***Corynebacterium mucifaciens*** (x2) | Staphylococcus capitis* |
| Staphylococcus capitis* | Staphylococcus hominis | Staphylococcus epidermidis | Roseomonas mucosa |
| ***Actinomyces neuii*** | Staphylococcus capitis | Staphylococcus hominis | |
| | | ***Roseomonas mucosa*** | |

Species listed in **bold italic** are those where the culture and sequencing data both show movement of the same unique arm species not present on the back. Species from $T_0$ that persist in the $T_{24}$ site (and remain absent at $Bb_{24}$ site) are annotated with a (*).

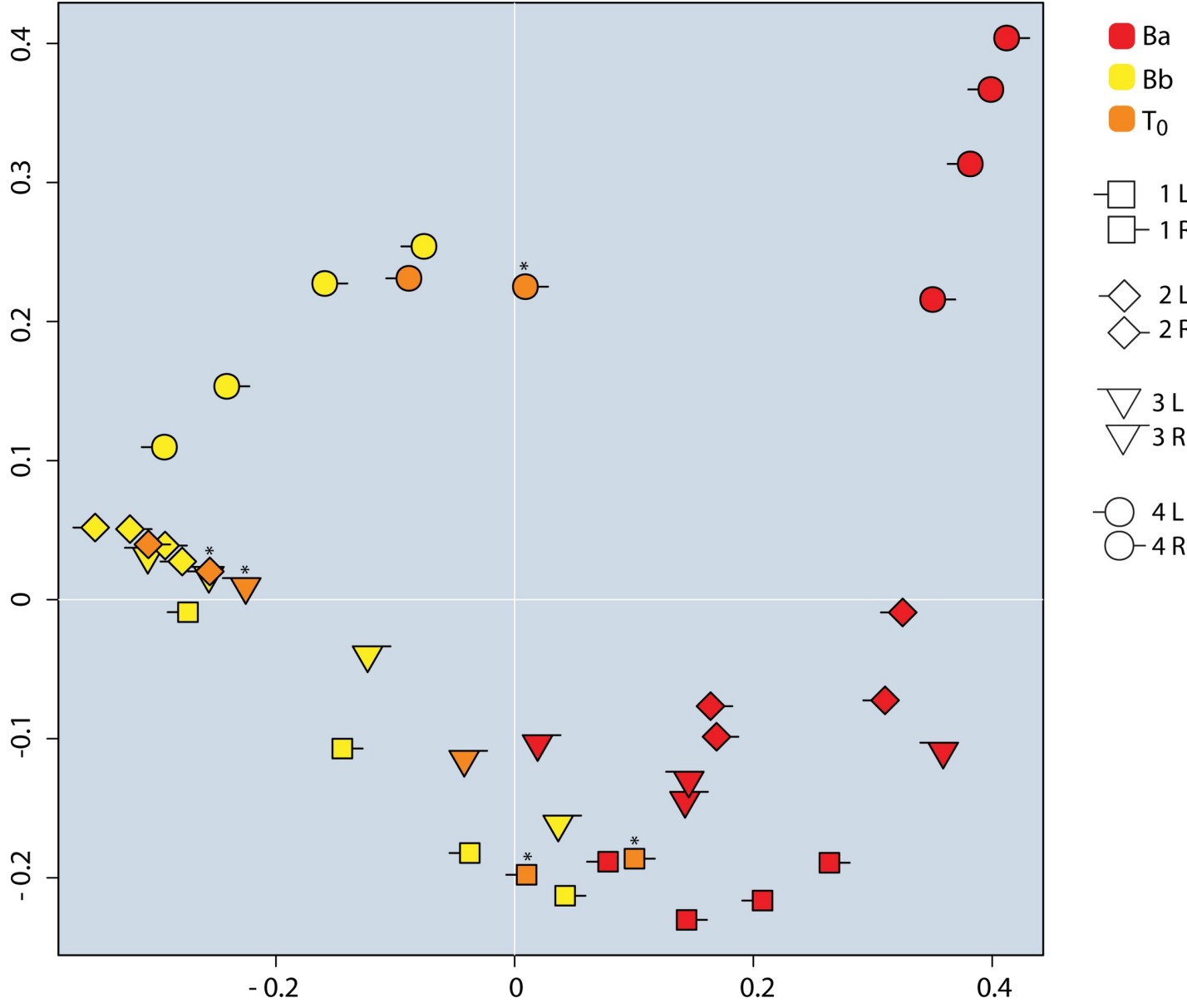

**Fig 5. PCoA Bray-Curtis Plot which relates the similarity in community structure between samples by plotting each sample as a point in two dimensions.** The shapes and color allow us to compare the baseline arm [Ba], baseline back [Bb], and recipient site samples [T$_0$] from each side of each individual. There is a trend in five of eight T$_0$ samples (orange), showing a shift "rightwards" of both of their corresponding back samples (same shape, only yellow), towards their corresponding arm samples (same shape, but red). These T$_0$ samples are denoted with a (*).

**Table 3. Viability of resuspended transfer pellet vs. standard skin swab, measured in colony forming units on blood agar (48 hrs).**

|  | Centrifuged transplant pellet (resuspended in Amies solution) | Standard skin swab (mixed in equivalent volume of Amies solution) |
|---|---|---|
| Replicate 1 | 1600 CFU | 1560 CFU |
| Replicate 2 | 1950 CFU | 1840 CFU |

we strictly interpret the persisting signal of unique arm bacterial DNA at $T_{24}$ (ignoring our pilot trials that showed the viability of transfer pellets, and our modest success at culturing unique arm species and colony morphotypes from the $T_0$ and $T_{24}$ samples), we cannot say whether it is merely residual from dead transferred bacteria 24 hours prior, whether there was a die-off from competition against resident bacteria, or whether the transferred bacteria didn't survive because they were poorly adapted to their new, sebaceous microenvironment. Another possibility is that growth and establishment of transferred bacteria takes more time to detect. Investigations in fecal microbiota transplantation show an incremental shift towards the donor microbiota signature that takes months to reach is fullest extent, with only partial engraftment detectable several days after transplant [26].

Within the context of previous literature, our finding that the antecubital fossa is significantly more diverse than the back is consistent with other descriptions of the skin microbiome; in one previous study where 20 distinct skin sites were ranked by evenness, the back was the least diverse, while the antecubital fossa was the 18th most diverse; when ranked by richness the back was the second least diverse and the antecubital fossa the 17th most diverse [1].

## Limitations

Limitations of our study begin with our difficulty culturing the bacterial species (unique to the arm) whose DNA we demonstrated moving with 16S sequencing. It was our intention to use culture to demonstrate the viability of this transplant "signal", but we did not effectively culture these organisms from baseline samples of the arm, nor the $T_0/T_{24}$ samples. Retrospectively, we were overly optimistic that we would be able to culture organisms that had been largely unrecognized prior to deep sequencing survey of the skin, even with our incorporation of special methods to grow gram negative species. We were also limited by the MALDI-TOF library, which has developed to identify clinically relevant isolates and was unable to define a number of the cultured isolates of commensal skin microbiota.

Another crucial limitation in our design was the lack of a control arm with heat-killed transfer samples. As an alternative to heat-treatment, we could have generated transfer pellets in ethanol at the centrifugation step instead of saline. If the non-viable transfer pellet (recovered at $T_0$ and $T_{24}$), showed less robust culture growth and a steeper drop-off in the persistence of unique DNA at 24 hours, we would have a much stronger claim that we had not just transferred a partial DNA signature, but viable organisms.

Other limitations of the study include the small number of participants (underpowered analysis), our focus on bacteria and exclusion of fungi and viruses, and the fact that our transplant is superficial, excluding the rich microbial habitats of appendageal structures (follicles and glands).

One unexpected finding was the number of species found exclusively in the $T_0$ samples. The $T_0$ samples showed a median of 45 unique species [range: 20,79] not found in the *Ba* or *Bb* samples of the same side of the study subject. We attribute this finding primarily to sample bias. Our sampling grids spanned an area from the antecubital fossa proper into the edge of volar forearm and the medial upper arm. Adding this slight geographical variability to the natural variability inherent in any two adjacent samples, we suspect that some of the bacteria in the pellet were not sampled from the arm at baseline, resulting in a number of species that appeared novel in the $T_0$ samples. A supporting fact is that many species unique to $T_0$ samples were found on the contralateral arm of the same study subject at baseline.

## Future directions and conclusions

With our pilot serving as a proof of concept that it is possible to transfer a partial DNA signature, the next step is to investigate the viability and colonization efficiency of transferred skin

microbiota between the same site of two different individuals. Using whole genome sequencing, we could follow strains of identical species from one individual to another. Without question, we would incorporate a heat or ethanol-treated control with each replicate. Longitudinal swabs, including at 24 hours and 240 hours, would give meaningful information about the persistence of a transplant, and by using the same body site between donor and recipient individuals, we can examine colonization efficiency without the confounding factor of a new microenvironment for transplanted bacteria.

We conclude that unenriched transfer of whole cutaneous microbiota is challenging, but our simple technique intended to move viable skin organisms from one site to another shows the first transfer of a partial DNA signature, and is worthy of further investigation and refinement. There still remain many questions in skin microbiota transplant including 1) whether a community of microbes, not any single, offer advantage in ensuring colonization at the recipient site, 2) whether there is one or a few particular organism(s) essential in restoring eubiosis, and thus skin health, and 3) how host immunity facilitates or inhibits colonization of a transplanted community.

## Supporting information

**S1 Photo. Supplementary photos.** (a) we placed wax/parchment paper over a template, wiped it with bleach, and constructed the grid over it with waterproof medical tape, which had been cut into strips (~0.63cm wide, which is ¼ the width of the tape); (b) the transplant grid was easily removed like a sticker from its backing and placed on a study subject for sampling. (TIF)

**S1 Dataset. Culture data.** Excel spreadsheet includes legend and data that document colony counts and subtyped cultures from each sample with their corresponding MALDI results. (XLSX)

## Acknowledgments

Funding sources included The National Center for Advancing Translational Sciences of the National Institutes of Health (Award Number UL1TR000423), as awarded by the Seattle Children's CCTR Pediatric Pilot Fund Program. The funders had no role in study design, data collection and interpretation, or the decision to submit the work for publication. The content is solely the responsibility of the authors and does not necessarily represent the official views of the NIH.

We would like to thank the infectious disease research group at Seattle Children's for their guidance and encouragement (particularly Dr. Matthew Kronman for his mentorship, in addition to Dr. Danielle Zerr, Amanda Adler, Dr. Alex Greninger, Dr. Scott Weissman, and Arianna Miles-Jay).

## Author Contributions

**Conceptualization:** Benji Perin, Amin Addetia.

**Data curation:** Benji Perin, Amin Addetia.

**Formal analysis:** Benji Perin.

**Funding acquisition:** Benji Perin, Amin Addetia, Xuan Qin.

**Investigation:** Benji Perin, Amin Addetia, Xuan Qin.

**Methodology:** Benji Perin, Amin Addetia, Xuan Qin.

**Project administration:** Benji Perin, Amin Addetia, Xuan Qin.

**Resources:** Benji Perin, Xuan Qin.

**Software:** Benji Perin.

**Supervision:** Benji Perin, Xuan Qin.

**Visualization:** Benji Perin, Amin Addetia.

**Writing – original draft:** Benji Perin.

**Writing – review & editing:** Benji Perin, Amin Addetia, Xuan Qin.

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
