## [Decision Letter · Decision Letter 0]

20 Sep 2019

PONE-D-19-23901

Transfer of skin microbiota between two dissimilar autologous microenvironments: a pilot study

PLOS ONE

Dear Dr. Perin,

Thank you for submitting your manuscript to PLOS ONE. After careful consideration, we feel that it has merit but does not fully meet PLOS ONE’s publication criteria as it currently stands. Therefore, we invite you to submit a revised version of the manuscript that addresses the points raised during the review process.

Please address all the reservations listed by the reviewers.

We would appreciate receiving your revised manuscript by Nov 04 2019 11:59PM. To enhance the reproducibility of your results, we recommend that if applicable you deposit your laboratory protocols in protocols.io, where a protocol can be assigned its own identifier (DOI) such that it can be cited independently in the future. For instructions see: http://journals.plos.org/plosone/s/submission-guidelines#loc-laboratory-protocols

We look forward to receiving your revised manuscript.

Kind regards,

Miroslav Blumenberg, PhD

Academic Editor

PLOS ONE

Journal Requirements:

2. We ask that you please state that you obtained written informed consent in your methods section, thank you for including this in your online ethics statement.

3. We note that 'Supplementary photos' includes an image of an individual.

As per the PLOS ONE policy (http://journals.plos.org/plosone/s/submission-guidelines#loc-human-subjects-research) on papers that include identifying, or potentially identifying, information, the individual(s) or parent(s)/guardian(s) must be informed of the terms of the PLOS open-access (CC-BY) license and provide specific permission for publication of these details under the terms of this license. Please download the Consent Form for Publication in a PLOS Journal (http://journals.plos.org/plosone/s/file?id=8ce6/plos-consent-form-english.pdf).

The signed consent form should not be submitted with the manuscript, but should be securely filed in the individual's case notes.

Please amend the methods section and ethics statement of the manuscript to explicitly state that the patient/participant has provided consent for publication: “The individual in this manuscript has given written informed consent (as outlined in PLOS consent form) to publish these case details”.

5. Please include your tables as part of your main manuscript and remove the individual files. Please note that supplementary tables should be uploaded as separate "supporting information" files.

6. Please include captions for your Supporting Information files at the end of your manuscript, and update any in-text citations to match accordingly. Please see our Supporting Information guidelines for more information: http://journals.plos.org/plosone/s/supporting-information

Reviewers' comments:

Reviewer's Responses to Questions

**Comments to the Author**

1. Is the manuscript technically sound, and do the data support the conclusions?

Reviewer #1: No

Reviewer #2: Yes

2. Has the statistical analysis been performed appropriately and rigorously? 

Reviewer #1: Yes

Reviewer #2: Yes

3. Have the authors made all data underlying the findings in their manuscript fully available?

Reviewer #1: Yes

Reviewer #2: Yes

4. Is the manuscript presented in an intelligible fashion and written in standard English?

Reviewer #1: Yes

Reviewer #2: Yes

5. Review Comments to the Author

Reviewer #1: The primary concern is a lack of agreeable definition for “transfer”. In this space the authors have assumed that the detection of DNA signature 0 or 24 hour after swabs constitutes transfer. As written, there is a possibility that they only transferred DNA, not viable organisms. In fact, the culture results in Table 3 argue strongly that they did not transfer a “viable community”, but at best transferred two coagulase negative Staph strains. Traditional culture techniques should not miss Staph epi – thus the lack of S. epi growth in the T24 cultures is concerning that no viable organisms were transferred in the first place.

A control could be performed swabbing the back with heat killed samples from the arm (i.e. culture Staph from the arm, heat kill it, then see if the bacterial 16S signature is present on the T0 and T24 samples). If the non-viable swab shows a less impressive transfer, the authors would have a much stronger claim. IE consider the direct non-viable culture swab a DNA-only, positive control to compare. An alternate approach would be to take the T0 swab and dip it in diluted EtOH prior to swabbing the back. If the 16S signature differed between the T24 and T24+EtOH group (versus a blank swab dipped in EtOH alone) the authors might have a stronger case that the T24 signature was from viable organisms.

Short of control samples, the entire paper (especially the abstract) requires that the language be amended to outline that they have transferred the DNA profile of the arm to the back. They have not provided the results needed to claim that they “can move viable skin organisms from one site to another”. There is no shame in showing that they were the first to transfer a partial DNA signature, but need to do further work to establish if they transferred viable organisms. But this paper claims multiple times to have transferred viable communities when they do not demonstrate that with their data.

This also needs to be discussed in detail in the limitations section, rather than the glancing comment contained now. As constructed, I feel that the overall language is misleading as it implies that swabs could be used from person A to person B to transfer viable organisms. This may be true, and is indeed “worthy of further investigation”, but without demonstrating colonization (even for as little as 24 hours), I cannot endorse this manuscript.

Minor concern:

In Fig 5, the authors state that the orange symbols – “the community has shifted towards the Ba samples (red) and away from the Bb samples (yellow)”. I don’t see this claim supported at all, can they provide mathematical support for this?

Reviewer #2: Perin et al present skin microbiome transplantation auto-transplantation from a site on the back to an arm site study in four patients. This work results interesting for the community but some aspects need to be clarified.

Major comments:

-It results unclear how it was measured the viability of the applied organisms on the transplant. It should be clarified how many live bacteria were transferred. Viability of the isolated microorganisms should be assayed and quantified

-Figures 3 and 5 should include the data of T24

Minor comments:

-What is the composition of the saline solution used for recovery?

-Tables are very hard to read. Version with improved resolution should be provided.

6. PLOS authors have the option to publish the peer review history of their article (what does this mean?). If published, this will include your full peer review and any attached files.

Reviewer #1: No

Reviewer #2: No

---

## [Author Response · Author response to Decision Letter 0]

6 Nov 2019

Thank you very much for your thorough review. We have responded to each comment in turn in our response to reviewers document.

---

## [Decision Letter · Decision Letter 1]

9 Dec 2019

Transfer of skin microbiota between two dissimilar autologous microenvironments: a pilot study

PONE-D-19-23901R1

Dear Dr. Perin,

We are pleased to inform you that your manuscript has been judged scientifically suitable for publication and will be formally accepted for publication once it complies with all outstanding technical requirements.

With kind regards,

Miroslav Blumenberg, PhD

Academic Editor

PLOS ONE

Additional Editor Comments (optional):

Reviewers' comments:

Reviewer's Responses to Questions

**Comments to the Author**

1. If the authors have adequately addressed your comments raised in a previous round of review and you feel that this manuscript is now acceptable for publication, you may indicate that here to bypass the “Comments to the Author” section, enter your conflict of interest statement in the “Confidential to Editor” section, and submit your "Accept" recommendation.

Reviewer #1: All comments have been addressed

Reviewer #2: All comments have been addressed

2. Is the manuscript technically sound, and do the data support the conclusions?

Reviewer #1: Yes

Reviewer #2: Yes

3. Has the statistical analysis been performed appropriately and rigorously? 

Reviewer #1: Yes

Reviewer #2: Yes

4. Have the authors made all data underlying the findings in their manuscript fully available?

Reviewer #1: Yes

Reviewer #2: Yes

5. Is the manuscript presented in an intelligible fashion and written in standard English?

Reviewer #1: Yes

Reviewer #2: Yes

6. Review Comments to the Author

Reviewer #1: All of my issues have been addressed. I think the paper is sound and will be of use for the research community - with the admitted limitations added I think they present their data in a fair light (showing the promise without over stating).

Reviewer #2: Authors addressed all the relevant issues pointed during review. Missing data from graphics has been clarified and Tables improved.

7. PLOS authors have the option to publish the peer review history of their article (what does this mean?). If published, this will include your full peer review and any attached files.

Reviewer #1: No

Reviewer #2: No

---

## [Editor Report · Acceptance letter]

16 Dec 2019

PONE-D-19-23901R1 

Transfer of skin microbiota between two dissimilar autologous microenvironments: a pilot study 

Dear Dr. Perin:

I am pleased to inform you that your manuscript has been deemed suitable for publication in PLOS ONE. Congratulations! Your manuscript is now with our production department. 

With kind regards,

on behalf of

Dr Miroslav Blumenberg 

Academic Editor

PLOS ONE